# Inhibition of circulating dipeptidyl-peptidase 3 restores cardiac function in a sepsis-induced model in rats: A proof of concept study

**Benjamin Deniau**[1,2], **Alice Blet**[1,2☯], **Karine Santos**[3], **Prabakar Vaittinada Ayar**[2,4,5], **Magali Genest**[2], **Mandy Kästorf**[3], **Malha Sadoune**[2], **Andreia de Sousa Jorge**[2], **Jane Lise Samuel**[2], **Nicolas Vodovar**[2], **Andreas Bergmann**[3], **Alexandre Mebazaa**[1,2,4]*, **Feriel Azibani**[2]

1 Department of Anesthesia, Burn and Critical Care, University Hospitals Saint-Louis- Lariboisière, AP-HP, Paris, France, 2 UMR-S 942, INSERM, MASCOT, Paris, France, 3 4TEEN4 Pharmaceuticals GmbH, Hennigsdorf, Germany, 4 Université de Paris, Paris, France, 5 Emergency Department, University Hospital of Beaujon, APHP, Clichy, France

☯ These authors contributed equally to this work.

* alexandre.mebazaa@aphp.fr

**Data Availability Statement:** All relevant data are within the manuscript and its Supporting Information files.

## Abstract

Sepsis is a global economic and health burden. Dipeptidyl peptidase 3 (DPP3) is elevated in the plasma of septic patients. The highest levels of circulating DPP3 (cDPP3) are found in non-survivor septic shock patients. The aim of this study was to evaluate the benefits of inhibiting cDPP3 by a specific antibody, Procizumab (PCZ), on cardiac function in an experimental model of sepsis, the caecal ligature and puncture (CLP) model. Rats were monitored by invasive blood pressure and echocardiography. Results are presented as mean ± SD, with $p < 0.05$ considered significant. PCZ rapidly restored left ventricular shortening fraction (from 39 ± 4% to 51 ± 2% before and 30 min after PCZ administration ($p = 0.004$)). Cardiac output and stroke volume were higher in the CLP + PCZ group when compared to the CLP + PBS group (152 ± 33 mL/min vs 97 ± 25 mL/min ($p = 0.0079$), and 0.5 ± 0.1 mL vs 0.3 ± 1.0 mL ($p = 0.009$), respectively) with a markedly reduced plasma DPP3 activity (138 ± 70 U/L in CLP + PCZ group versus 735 ± 255 U/L ($p = 0.048$) in the CLP + PBS group). Of note, PCZ rapidly reduced oxidative stress in the heart of the CLP + PCZ group when compared to those of the CLP + PBS group (13.3 ± 8.2 vs 6.2 ± 2.5 UI, $p = 0.005$, 120 min after administration, respectively). Our study demonstrates that inhibition of cDPP3 by PCZ restored altered cardiac function during sepsis in rats.

## Introduction

Sepsis is as a life-threatening organ dysfunction caused by a dysregulated host response to infection with major impact worldwide [1]. Septic shock, the deadliest form of sepsis, is characterized by low systemic blood pressure despite fluid resuscitation, organ dysfunction and high mortality, especially when accompanied by myocardial depression [2,3]. The precise

**Funding:** The study was mainly supported by Inserm, Université of Paris and Assistance Publique- Hôpitaux de Paris, Paris France. 4TEEN4 Pharmaceuticals GmbH provided vials blinded for placebo or Procizumab, measured DPP3 activity and gave an unrestricted research grant to MASCOT research group that allowed salary support for one co-author (BD). 4TEEN4 Pharmaceuticals GmbH did not have a role in design, data collection and analysis, decision to publish or preparation of the manuscript. Specific roles of the authors are described in the "authors contribution section".

**Competing interests:** AM reports personal fees from Orion, Servier, Otsuka, Philips, Sanofi, Adrenomed, Epygon and Fire 1 and grants and personal fees from 4TEEN4 Pharmaceuticals GmbH, Abbott and Sphingotec. BD and AB were invited to meetings in Hennigsdorf by 4TEEN4 Pharmaceuticals GmbH. AB is CEO and shareholder at 4TEEN4 Pharmaceuticals GmbH. KS and MK are employees of 4TEEN4 Pharmaceuticals GmbH. Other authors declared no potential conflicts of interest with respect to the research authorship and/or publication of this article. 4TEEN4 Pharmaceuticals GmbH holds patent rights on the DPP3 biomarker and humanized antibody Procizumab. This does not alter our adherence to PLOS ONE policies on sharing data and materials.

mechanisms of organ dysfunction in sepsis still remain elusive, especially the induced-cardiac dysfunction [4,5].

Dipeptidyl peptidase 3 (DPP3) is a ubiquitously expressed intracellular enzyme that is involved in the cleavage of small peptides [6]. Exact roles of DPP3 in the heart and especially in cardiac cells are not completely understood. We recently showed that circulating DPP3 (cDPP3) was elevated in cardiogenic shock patients and that high levels of cDPP3 were associated with altered hemodynamic and poor outcomes [7]. Experimental studies showed that bolus administration of natively purified human DPP3 provoked a rapid and marked deterioration of heart contractile function in healthy mice [7]. Furthermore, inhibition of cDPP3 by its specific antibody, Procizumab (PCZ), promptly restored cardiac contractility and decreased myocardial oxidative stress in a mouse model of acute cardiac stress, in which cDPP3 is increased [7].

In a preliminary study of a cohort of septic patients, those patients with septic shock showed higher levels of cDPP3 compared to patients with severe sepsis [8]. We hypothesised that cDPP3 could have a role in the pathophysiology of cardiac dysfunction during septic shock. We, therefore, evaluated the effect of DPP3 inhibition by PCZ in a rat model of cardiac dysfunction induced by septic shock.

# Material and methods

## Sepsis model

Three-month-old male Wistar rats weighing 350–450 g (Janvier, St Berthevin, France) were used. All experiments were conducted in accordance with the National and European Institutes of Health Guidelines for the use of laboratory rats and were approved by the research ethics committee of Paris University (protocol number S140 #9385) and complied to ARRIVE guidelines and all efforts were made to minimize suffering. Septic shock and hemodynamic monitoring were done as previously described [9,10]. All rats were anaesthetized using an intraperitoneal injection of ketamine hydrochloride (90 mg/kg) and xylazine (9 mg/kg). Polymicrobial sepsis was provoked by cecal ligation and puncture (CLP) as previously described [10]. Briefly, a ventral midline incision (1 cm) was made to allow exteriorization of the caecum. The caecum was then ligated just below the ileocecal valve and punctured once with an 18-gauge needle. The abdominal cavity was closed in two layers, and rats were given fluid resuscitation (3 mL/100 g of body weight) of saline, injected subcutaneously). A sham operation was carried out by isolating the caecum with neither ligation nor puncture (sham + PBS group). Intraperitoneal injection of 75 μg/kg of buprenorphine was used for analgesic purposes in the preoperative period. Pain was evaluated by using visual analogue scale score sheet after CLP surgery, and analgesia was adapted by using buprenorphine if needed. 16 hours later, after general anaesthesia with ketamine hydrochloride (90 mg/kg) and xylazine (9mg/kg), rats were placed in the supine position and intubated with a 16 G catheter and ventilated using a rodent ventilator with the respiratory rate set at $53.5 \times weight^{-0.26}$ and the tidal volume set at $6.2 \times weight^{1.01}$ [9]. Rectal temperature was maintained throughout the protocol at 37–37.5˚C by a heating mat. Catheters were inserted into the left jugular vein to administer treatment or placebo and into the right carotid artery for invasive blood pressure monitoring.

Based on the SEPSIS-3 consensus definition of septic shock [1], we only included rats with mean blood pressure (MBP) below 65 mmHg. To ensure cardiac dysfunction, we included only rats with a left ventricular shortening fraction (LVSF) below 45%. We plan to have 5 animals alive at the end of the 120 min hemodynamic study. To do so, we initially blinded five animals in each group (CLP + PBS and CLP + PCZ). Given the high mortality before the end of the 120-minute follow-up in the CLP + PBS group, we had to include additional animals to

achieve 5 animals surviving at 120 minutes in this group. After a bolus injection of 1.5 mL PBS or PCZ (concentration of 1 mg/ml) through the jugular vein in 5 min, which is equivalent as a fluid resuscitation, PBS (2 mL) and PCZ were continuously infused with an electric syringe (2 mL at the concentration of 0.42 mg/mL) during the 120 minutes of follow-up. The experimental protocol is summarized in Fig 1A.

## Hemodynamics and cardiac function monitoring

Monitoring of hemodynamics and cardiac function was performed as previously described [9,11]. Echocardiography parameters were measured by transthoracic echocardiography (TTE) using Vivid 7 system (General Electrics, USA) equipped with a 14-MHz linear transducer for rats, before randomization (16 hours after CLP or sham procedure) and every 30 min during the 120 minutes of the protocol (Fig 1A). All acquisitions were recorded digitally and stored for subsequent off-line analysis [12]. Invasive blood pressure measurements, by using arterial catheter in carotid artery and recorded by Biopac® system and AcqKnowledge® 5.0 software (California, USA) for data acquisition, were done before randomization and every 30 minutes after the beginning of therapy (PBS or PCZ injection) during the 120 minutes of the protocol.

## Inhibition of DPP3 activity by PCZ

Generation, screening and development of humanized anti-DPP3 antibody PCZ was previously described [7]. In addition, the humanized DPP3-inhibiting antibody Procizumab binds to a conserved, surface exposed loop in proximity to the active site. Crystal structures of human DPP3 in apo and substrate-bound forms show that the enzyme undergoes a large conformational change between the open (apo) state and the closed (substrate-bound state). Since the epitope targeted by Procizumab remains exposed in the surface in the open and closed states, it is expected that the antibody binds the enzyme regardless of its conformation and sterically blocks the transition between DPP3 catalytic states, therefore, inhibiting enzyme activity.

## Anatomical examination and tissue preparation

At the end of the protocol, the rats were sacrificed by lethal anaesthesia with an injection of 40mg/kg of pentobarbital solution (CEVA, Libourne, France). The heart was transversally divided into two parts: the base part was embedded into Tissue-Tek optimal cutting temperature (OCT) compound (VWR Chemicals, Jeuven, Belgium) and frozen into liquid nitrogen pre-cooled isopentane; the apex was snap-frozen in liquid nitrogen. All samples were stored at −80˚C for further analyses. Heparin blood was collected from the left carotid artery at the end of the experiment, centrifuged at 3500 rpm for 15 min at 4˚C, and plasma was stored at -80˚C.

## Plasma DPP3 activity

Plasma DPP3 activity was measured using the soluble activity assay (SAA) with a fluorogenic substrate (15). Briefly, 10 µl of rat heparin-plasma were incubated with 90 µl of a substrate reagent solution (50 mM Tris/HCl, pH 7.8 (25˚C), 0.125% Triton X-100, 100 µM Arg2-β-napthylamide (Arg2-βNA, Bachem AG) for 1 h at 37˚C in black 96-well microtiter plates. Fluorescence of the cleaved product βNA was detected at 420 nm using the Twinkle LB 970 fluorometer (Berthold Technologies GmbH).

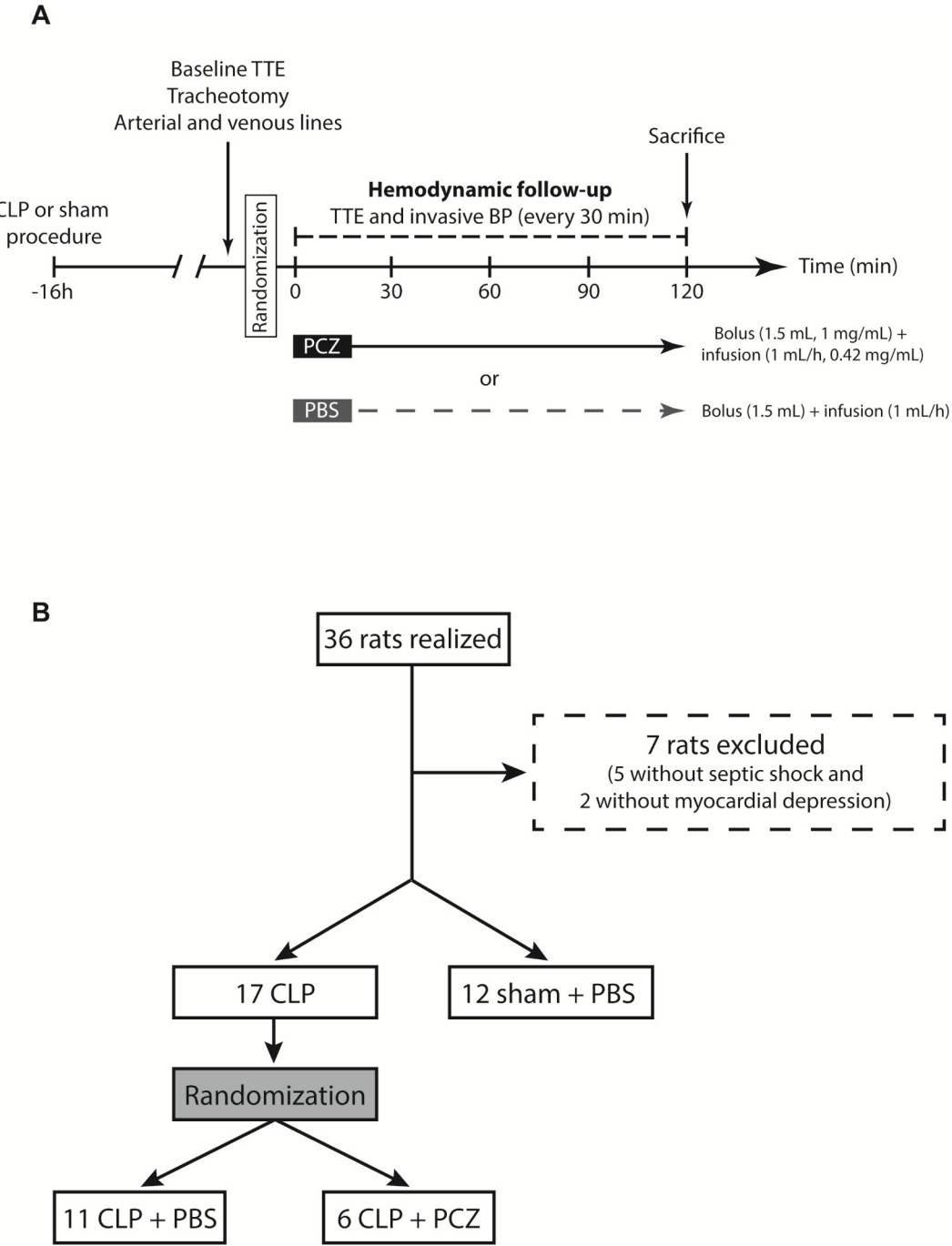

**Fig 1. Design and flow chart of the study.** (A) Schematic representation of the preclinical experiment in rats, including treatments and hemodynamic assessment (BP: blood pressure, CLP: cecal ligation and puncture, PBS: phosphate buffered saline, PCZ: Procizumab, TTE: transthoracic echocardiography). (B) Flowchart of rat's randomization.

## Gene expression analyses

Total RNA were isolated from tissues using the RNeasy Mini Kit® (Qiagen, Courtaboeuf, France) according to the manufacturer's instructions and reverse transcribed using Quanti-Tect® Reverse Transcription (Qiagen, Courtaboeuf, France). Subsequently, real-time

quantitative polymerase chain reaction was performed on a LightCycler96 (Roche Diagnostics, Meylan, France) using the FastStart Essential DNA Green Master® (Roche Diagnostics, Meylan, France). Transcripts levels for genes of interest were normalized to that of Glyceraldehyde-3-phosphate dehydrogenase (GAPDH) and expressed as the relative change compared to the control samples. The sequences of the primers used are reported in S1 Table.

### Dihydroethidium staining

Dihydroethidium staining (DHE, Sigma-Aldrich) was performed on 7 μm myocardial cross-cryosection as previously described [7,9].

### Statistical analysis

Data are expressed as mean ± SD. Normality was assessed using the Shapiro–Wilk test. For two-group comparisons, Wilcoxon signed-rank test or Wilcoxon rank-sum test were used as appropriate. Comparison between more than two groups was performed by one-way ANOVA, Kruskal–Wallis test or repeated-measures ANOVA followed by Dunn's multiple comparison test, as appropriate. A Kaplan-Meier curve and log-rank survival were done with a 95% confidence interval to evaluate survival. A p-value $<0.05$ was considered statistically significant. Statistical analysis was realized using R software (R Core Team, 2014) and figures were produced using ggplot2 (Wickham 2009).

## Results

### PCZ decreased plasmatic DPP3 activity and myocardial DPP3 transcription in septic rats

Sixteen hours after the CLP or sham procedure, clinical signs of sepsis (reduced motor activity, lethargy, shivering, piloerection and hunched posture) were only present in CLP and not in sham + PBS rats. Furthermore, post-mortem examination of the abdominal cavity of all CLP rats showed varying degrees of peritonitis with a grey-black dilated caecum and purulent and malodorous peritoneal fluid. Overall, thirty-six rats were enrolled in the study: 11 in the CLP + PBS group, 6 in the CLP + PCZ group, and 12 in the sham + PBS group (Fig 1B). Baseline characteristics of the sham and CLP rats after randomisation are summarized in Table 1.

Interestingly, we found that the CLP + PBS rats had a higher cDPP3 plasma activity than the sham + PBS rats at 120 min (735 ± 255 UI/L vs 94 ± 41 UI/L, p = 0.0063 respectively) (Fig 2A). Similarly, *DPP3* transcripts were up-regulated in the heart of the CLP + PBS rats

**Table 1. Hemodynamics and cardiac parameters in sham and CLP rats before randomisation (i.e. before PBS or PCZ injection).**

|  | Sham | CLP before randomization | p |
|---|---|---|---|
|  | n = 12 | n = 17 |  |
| LVSF (mean ± SD) | 54 ± 7% | 39 ± 4% | $<0.001$ |
| SBP (mean ± SD) | 99 ± 11 mmHg | 79 ± 10 mmHg | $<0.001$ |
| DBP (mean ± SD) | 70 ± 12 mmHg | 46 ± 10 mmHg | $<0.001$ |
| MBP (mean ± SD) | 79 ± 11 mmHg | 57 ± 8 mmHg | $<0.001$ |
| HR (mean ± SD) | 321 ± 77 bpm | 341 ± 62 bpm | 0.428 |
| CO (mean ± SD) | 113 ± 33 mL/min | 99 ± 26 mL/min | 0.231 |
| SV (mean ± SD) | 0.4 ± 0.1 mL | 0.3 ± 0.1 mL | 0.064 |

Values are expressed as mean ± SD. Comparisons were done by using the Wilcoxon rank-sum test. (CO: cardiac output, DBP: diastolic blood pressure, HR: heart rate, LVSF: left ventricular shortening fraction, MBP: mean blood pressure, SBP: systolic blood pressure, SV: stroke volume).

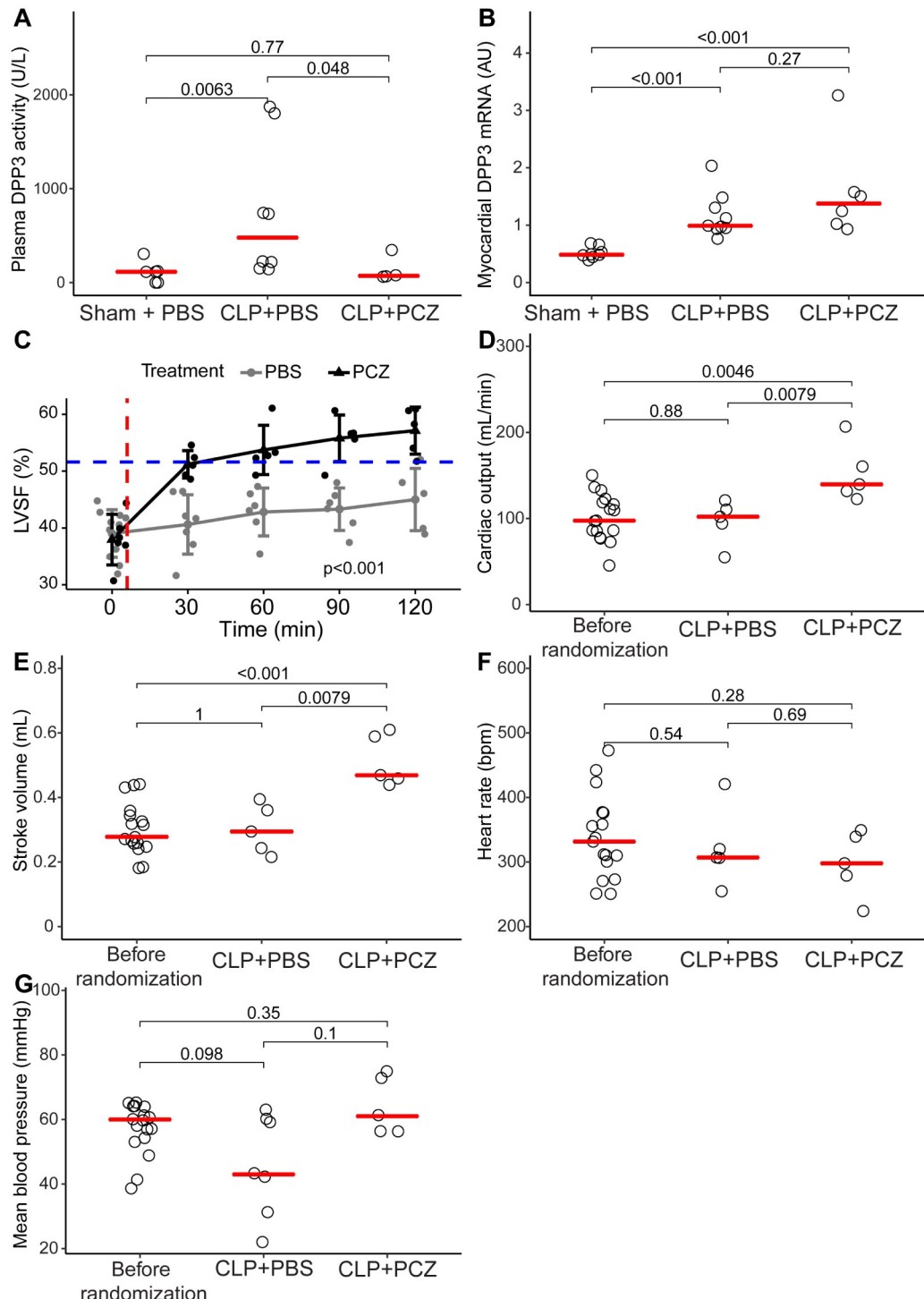

**Fig 2. PCZ injection decreased DPP3 plasmatic activity and improved hemodynamics and cardiac function during sepsis.** (A) Plasmatic DPP3 activity in sham + PBS, CLP + PBS and CLP + PCZ groups. Comparisons were made using Wilcoxon rank-sum test. (B) Myocardial qPCR of DPP3 of sham + PBS, CLP + PBS and CLP + PCZ rats. Comparisons were made by Wilcoxon rank-sum test. (C) Schematic representation of the evolution of the left ventricular shortening fraction in time in sham + PBS (blue dashed line), CLP + PBS (grey line) and CLP + PCZ groups (black line) (p<0.05 using repeated measures ANOVA). Red dashed line corresponds to the moment of bolus injection followed by infusion. (D) Evolution of the cardiac output, (E) stroke volume, (F) heart rate and (G) mean blood pressure before randomization and 120 min after therapy by PBS or PCZ injection in septic rats. Comparisons were realized by using Wilcoxon rank-sum test. (bpm: beats per minutes, CLP: cecal ligature and puncture).

compared to the sham + PBS group (1.17 ± 0.40 AU in CLP + PBS compared to 0.52 ± 0.10 in the sham + PBS group, p<0.001) (Fig 2B). Plasma cDPP3 activity was lower at 120 min in the PCZ group when compared to the PBS group (138 ± 139 U/L vs 735 ± 722 U/L, p = 0.048) (Fig 2A), without modifying the myocardial transcription of *DPP3* (1.59 ± 0.35 vs 1.17.± 0.13 in CLP + PBS group, p = 0.27) (Fig 2B).

### PCZ promptly restored cardiac systolic dysfunction in septic rats

PCZ administration rapidly restored LVSF from 39 ± 4% to 51 ± 2% within 30 min of PCZ initiation (p = 0.004). LVSF was higher in CLP + PCZ group when compared to CLP + PBS group (p<0.001) (Fig 2C). The LVSF of the PBS rats remained unchanged (p = 0.556; Fig 2C). At the end of the protocol–i.e. 120 minutes after the initiation of PCZ or PBS—administration of PCZ was associated with a greater CO (Fig 2D) and SV (Fig 2E) compared to the rats receiving PBS. In contrast, heart rate (HR) and MBP measured at 120 minutes were similar in all CLP groups (Fig 2F and 2G). Finally, the survival rate was higher at 120 min in the PCZ group compared to the PBS group (83% versus 63%, p = 0.0026, log-rank test) (S1 Fig). Of note, LVSF and other hemodynamic parameters remained stable throughout the duration of the experiment in the sham + PBS group (S2 Fig).

At the cardiac level, PCZ administration rapidly reduced oxidative stress in the heart, measured by DHE staining, compared to PBS administration (13.3 ± 8.2 vs 6.2 ± 2.5 UI, p = 0.0047, 120 min after PBS or PCZ injection, respectively; Fig 3A and 3B). However, the PCZ effect was not accompanied by any change in the myocardial *HO-1* and *NQO1* expression, two oxidative stress-induced genes (7.90 ± 4.38 AU CLP + PCZ vs 5.42 ± 2.03 AU CLP + PBS p = 0.33 and 5.21 ± 2.33 AU CLP + PCZ vs 4.49 ± 1.91 CLP+PBS, p = 0.61) (Fig 3C and 3D).

## Discussion

Our study demonstrates that inhibition of cDPP3 by PCZ restored cardiac contractility in septic rats. These data indicate that cDPP3 plays an important part in sepsis-induced myocardial depression and suggest PCZ as a promising therapeutic option in this context. Fig 4 resumes the main findings of our work.

The pathophysiology of sepsis-induced myocardial depression is still not fully understood. Parrillo et al. have described for the first time a "myocardial depressant factor" [13]. Since this hypothesis was drawn, several factors (including cytokines) were described to be associated with myocardial depression during sepsis or other acute conditions [14–19]. Other authors hypothesized that the down-regulation or uncoupling of adrenergic receptors induced by sepsis can explain the absence of response to endogenous and exogenous catecholamines, leading to septic shock [20]. Furthermore, an increased oxidative stress level may be implicated in triggering myocardial dysfunction during sepsis [21–23]. Finally, alteration of the angiotensin II pathway is another potential, but poorly studied, mechanism implicated in septic cardiomyopathy. The RAAS pathway is clearly modified during sepsis [20] with altered angiotensin II levels and reduced angiotensin II sensitivity, via the reduction or the uncoupling of angiotensin II receptors, and the consequent deterioration of its positive inotropic effects [20,24–27]. In the present study, by using a sepsis-induced model in rat, we demonstrated that blocking cDPP3 with PCZ resulted in prompt and marked cardiac contractile benefits. The present data reinforce cDPP3 as a potential myocardial depressant factor together with previous benefits of PCZ in cardiogenic shock [7]. Furthermore, the elevation of plasma cDPP3 could reduce angiotensin II levels by cleavage into angiotensin IV [6,28], as we previously observed in mouse and human plasma samples spiked with natively purified DPP3 [7]. Thus, the

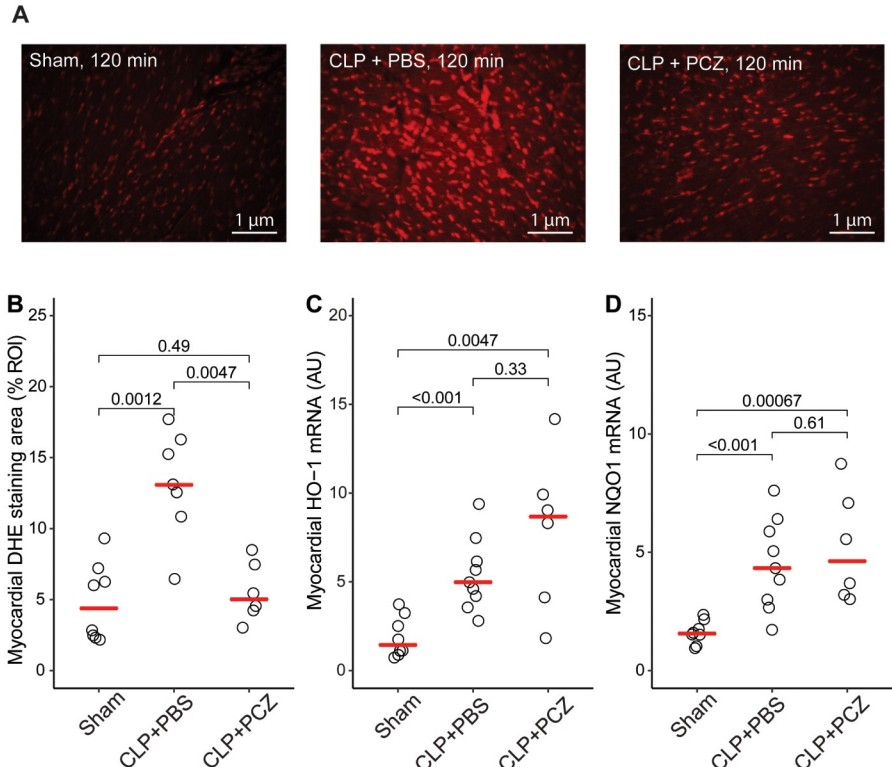

**Fig 3. PCZ injection improved myocardial oxidative stress.** Representative images (A) and quantification (B) of DHE staining from myocardial sections of sham + PBS, CLP + PBS and CLP + PCZ rat groups. Myocardial mRNA expression of *HO-1* and *NQO-1* in sham + PBS, CLP + PBS and CLP + PCZ groups. Comparisons were made by Wilcoxon rank-sum test. (DHE: dihydroethidium, *HO-1*: heme oxygenase 1, *NQO-1*: NAD(P)H dehydrogenase (quinone 1)).

restoration of the cardiac function in septic rats by blocking DPP3 could result from the potentiation of angiotensin II that would be normally inactivated. Altogether, these data strongly suggest that cDPP3 could be a sought-after myocardial depressing factor, which would act by decreasing the angiotensin II signalling pathway when increased in septic shock.

Our study has some limitations. First, the duration of the protocol is short as we designed this proof of principle study based on the short life span of the septic rats (18 to 20 hours post-surgery). Second, septic rats have not been fully resuscitated, particularly, we did not treat the rats with antibiotics. However, given the duration of the protocol, the administration of antibiotics is not expected to provide any improvement on the rat status. Along the same line, rats did not receive any exogenous catecholamine as the aim was to evaluate the hemodynamic benefit of cDPP3 inhibition. Nevertheless, further studies are warranted to evaluate the benefit of PCZ in fully resuscitated rats, including probabilistic antibiotherapy and catecholamine administration. Third, the mechanism of action of DPP3 and its inhibition by PCZ in septic cardiomyopathy is currently poorly understood. However, the involvement of blood pressure in the improvement of cardiac function upon PCZ treatment seems to be limited,. Further studies are needed to determine PCZ's mechanism of action in heart function improvement during septic cardiomyopathy. Fourth, DPP3 activity was only measured in plasma since the antibody can only bind and inhibit DPP3 activity in plasma and not inside cells, e.g. cardiac cells. Fifth, we do not assess sex differences responses to PCZ treatment as only male mice

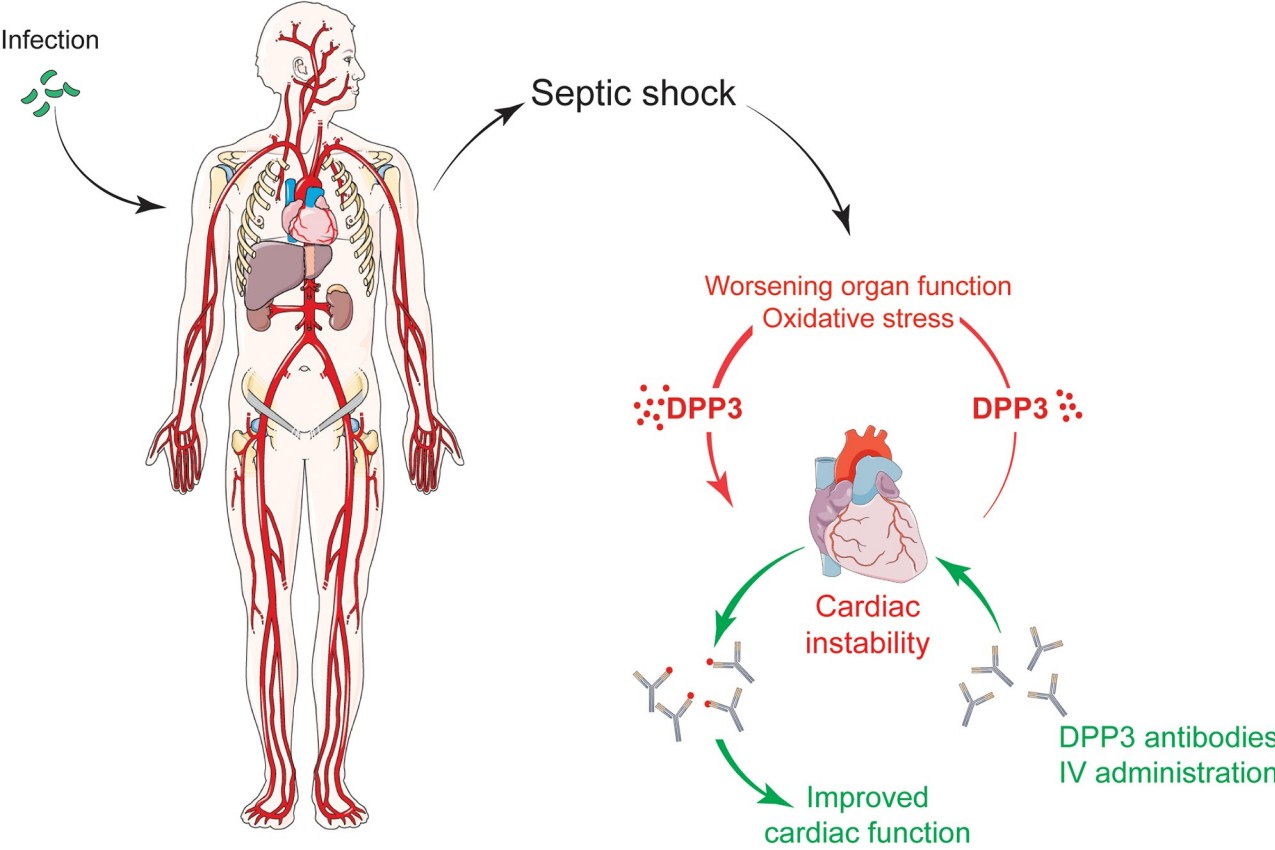

**Fig 4. cDPP3 is implicated in septic myocardial depression.** Its inhibition by a specific antibody, Procizumab, could be a tool to restore impaired cardiac function. Fig was built using https://smart.servier.com.

were included in this preclinical study. This is a common bias in experimental studies and more studies are needed to fill this gap.

We previously showed that cDPP3 is involved in acute heart failure in mice [7]. Our study extends the role of cDPP3 to septic cardiomyopathy. Additionally, inhibition of high levels of cDPP3 by PCZ in septic rats restored cardiac contraction and improved survival, suggesting PCZ as a possible therapeutic option in patients with septic shock, an indication where limited treatment efficacy has been observed [29]. The use of PCZ could possibly limit the use of high doses of inotropes during septic shock treatment, which has been associated with deleterious effects [30,31].

## Supporting information

**S1 Fig. Survival curve 120 min after PBS or PCZ injection.** CLP: cecal ligation and puncture, PBS: phosphate buffered saline, PCZ: Procizumab. Log rank test was used.
(TIF)

**S2 Fig. Hemodynamics and cardiac parameters remain stable in sham rats.** Schematic representation of the evolution of the left ventricular shortening fraction (A), mean blood pressure (B), cardiac output (C), stoke volume (D) and heart rate (E) 16 hours after sham surgical procedure. Comparisons were made by using repeated measures of ANOVA.
(TIF)

**S1 Table. Primers used for q*PCR*.**
(DOCX)

**S1 Data.**
(XLTX)

## Author Contributions

**Conceptualization:** Benjamin Deniau, Alice Blet, Karine Santos, Jane Lise Samuel, Nicolas Vodovar, Alexandre Mebazaa, Feriel Azibani.

**Data curation:** Benjamin Deniau.

**Formal analysis:** Benjamin Deniau, Karine Santos, Jane Lise Samuel, Nicolas Vodovar, Alexandre Mebazaa, Feriel Azibani.

**Funding acquisition:** Karine Santos, Andreas Bergmann.

**Investigation:** Benjamin Deniau, Karine Santos, Jane Lise Samuel, Feriel Azibani.

**Methodology:** Benjamin Deniau, Alice Blet, Karine Santos, Prabakar Vaittinada Ayar, Magali Genest, Mandy Kästorf, Malha Sadoune, Andreia de Sousa Jorge, Jane Lise Samuel, Nicolas Vodovar, Alexandre Mebazaa, Feriel Azibani.

**Project administration:** Karine Santos.

**Supervision:** Alice Blet, Karine Santos, Jane Lise Samuel.

**Validation:** Alice Blet, Karine Santos, Jane Lise Samuel, Alexandre Mebazaa, Feriel Azibani.

**Visualization:** Benjamin Deniau, Alice Blet, Karine Santos, Jane Lise Samuel, Alexandre Mebazaa, Feriel Azibani.

**Writing – original draft:** Benjamin Deniau, Karine Santos, Alexandre Mebazaa, Feriel Azibani.

**Writing – review & editing:** Benjamin Deniau, Karine Santos, Nicolas Vodovar, Alexandre Mebazaa, Feriel Azibani.

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
