## [Decision Letter · Decision Letter 0]

5 Jun 2020

PONE-D-20-10991

Inhibition of circulating dipeptidyl-peptidase 3 restores cardiac function in a sepsis-induced model in rats: a proof of concept study

PLOS ONE

Dear Dr. Deniau,

Thank you for submitting your manuscript to PLOS ONE. After careful consideration, we feel that it has merit but does not fully meet PLOS ONE’s publication criteria as it currently stands. Therefore, we invite you to submit a revised version of the manuscript that addresses the points raised during the review process.

All issues raised by expert reviewers are required.

We look forward to receiving your revised manuscript.

Kind regards,

Vincenzo Lionetti, M.D., PhD

Academic Editor

PLOS ONE

Journal Requirements:

2. To comply with PLOS ONE submissions requirements, in your Methods section, please provide additional information on the animal research and ensure you have included details on efforts to alleviate suffering.

'This study was partially supported by 4TEEN4 Pharmaceuticals GmbH.'

We note that one or more of the authors have an affiliation to the commercial funders of this research study: 4TEEN4 Pharmaceuticals GmbH

5. Please upload a copy of Figure 4, to which you refer in your text on page 12. If the figure is no longer to be included as part of the submission please remove all reference to it within the text.

Reviewers' comments:

Reviewer's Responses to Questions

**Comments to the Author**

1. Is the manuscript technically sound, and do the data support the conclusions?

Reviewer #1: Yes

Reviewer #2: Yes

2. Has the statistical analysis been performed appropriately and rigorously? 

Reviewer #1: Yes

Reviewer #2: Yes

3. Have the authors made all data underlying the findings in their manuscript fully available?

Reviewer #1: Yes

Reviewer #2: Yes

4. Is the manuscript presented in an intelligible fashion and written in standard English?

Reviewer #1: Yes

Reviewer #2: Yes

5. Review Comments to the Author

Reviewer #1: The authors previously showed that dipeptidyl peptidase 3 (DPP3) is a myocardial depressant substance. In this study, they used a model of septic shock induced by peritonitis to show that inhibition of cDPP3 by its specific antibody, procizumab (PCZ), improves the hemodynamic status.

The study is interesting and well performed.

Specific comments

1. How come the randomization was unbalanced? How was the randomization performed?

2. The effects on arterial pressure are hardly considered. It seems that the major effect of PCZ was an increase in arterial pressure, and this could in turn improve myocardial function.

3. Discussion: the suggestion that DPP3 could be the long-searched myocardial depressing factor is too much. The search for the MDF has been abandoned for a long time: It is now well accepted that there are many myocardial depressant substances.

Minor general comment

The text needs language editing: as an example,

-respectively appears before the data

-‘septic shock was performed…’

Reviewer #2: The manuscript by Deniau and colleagues demonstrates the potential role of DPP3 in cardiac dysfunction in an experimental model of sepsis. The manuscript is important, however I have number of critiques/suggestions to improve the manuscript detailed below.

MAJOR

1) What is the role of DPP3 in the heart? Is there any relationship between DPP3 and DPP4?

2) Why the antibody inhibits DPP3 activity? Is the epitope close to the active site of the enzyme?

3) Why DPP3 activity was measured only in the plasma but not in the heart of the animals? Is DPP3 cleaved by a sheddase? Is this sheddase upregulated in sepsis?

4) Very little is shown and discussed regarding the potential mechanisms by which DPP3 inhibition improves cardiac function in the setting of sepsis.

5) Was the lack of variation in heart rate among the three groups of rats expected?

MINOR

Please, replace angiotensin 2 by angiotensin II or Ang II.

6. PLOS authors have the option to publish the peer review history of their article (what does this mean?). If published, this will include your full peer review and any attached files.

Reviewer #1: Yes: Jean-Louis Vincent

Reviewer #2: No

---

## [Author Response · Author response to Decision Letter 0]

20 Jul 2020

Reviewer #1: The authors previously showed that dipeptidyl peptidase 3 (DPP3) is a myocardial depressant substance. In this study, they used a model of septic shock induced by peritonitis to show that inhibition of cDPP3 by its specific antibody, procizumab (PCZ), improves the hemodynamic status.

The study is interesting and well performed.

Specific comments

1. How come the randomization was unbalanced? How was the randomization performed?

Response: According to the ARRIVE guidelines, and to respect the 3R rule, number of tested animals was limited to the minimum required number. Additionally, for statistical reasons, a number of 5 animals per group at 120 min was needed. Because of the high mortality rate observed in the CLP + PBS group, we had to increase the number of animals in this group to reach 5 animals at the 120 min time point. 

The randomization of CLP rats in PBS or PCZ groups was blinded in order to obtain a minimum of 5 rats by group. The manuscript has been modified accordingly. 

Material and methods section, Page 3, Line 59 “We plan to have 5 animals alive at the end of the 120 min hemodynamic study. To do so, we initially blinded five animals in each group (CLP + PBS and CLP + PCZ). Given the high mortality before the end of the 120-minute follow-up in the CLP + PBS group, we had to include additional animals to achieve 5 animals surviving at 120 minutes in this group.)”.

2. The effects on arterial pressure are hardly considered. It seems that the major effect of PCZ was an increase in arterial pressure, and this could in turn improve myocardial function.

Response: We thank the reviewer for this interesting comment. As shown in Fig 2G, no effect was observed on mean blood pressure 120 min after PCZ administration. Thus, involvement of blood pressure in the improvement of cardiac function seems to be limited. Further studies are needed to determine PCZ’s mechanism of action in cardiac improvement during septic cardiomyopathy.

The manuscript was modified accordingly in the limitation section.

Discussion section, Page 10, line 217: “Third, the mechanism of action of DPP3 and its blockage by PCZ in septic cardiomyopathy is currently poorly understood. However, involvement of blood pressure in the improvement of cardiac function upon PCZ treatment seems limited. Further studies are needed to determine PCZ’s mechanism of action in cardiac improvement during septic cardiomyopathy”

 

3. Discussion: the suggestion that DPP3 could be the long-searched myocardial depressing factor is too much. The search for the MDF has been abandoned for a long time: It is now well accepted that there are many myocardial depressant substances.

Response: We thank the reviewer for this comment and we modified the manuscript accordingly. 

Discussion section, Page 11, line 224 “We previously showed that cDPP3 is involved in acute heart failure in mice [7]. Our study extends the role of cDPP3 to septic cardiomyopathy”.

4. Minor general comment

The text needs language editing: as an example,

-respectively appears before the data

-‘septic shock was performed…’

Response: We thank the reviewer for this comment. The manuscript has been grammatically reviewed and modified accordingly. 

Reviewer #2: The manuscript by Deniau and colleagues demonstrates the potential role of DPP3 in cardiac dysfunction in an experimental model of sepsis. The manuscript is important, however I have number of critiques/suggestions to improve the manuscript detailed below.

MAJOR

1) What is the role of DPP3 in the heart? Is there any relationship between DPP3 and DPP4?

Response: We thank the reviewer for this interesting comment. Currently, exact roles of DPP3 in the heart and especially in cardiac cells are not completely understood. DPP3 is a ubiquitous cytosolic enzyme involved in oxidative stress metabolism. Extracellular, circulating DPP3 has a role in the cleavage of cardiovascular peptides (notably Ang II and IV) as well as in pain and inflammatory modulation. We previously demonstrated that cDPP3 is involved in myocardial depression when injected in healthy mice (Deniau et al. European Journal of Heart Failure 2020). Upon inflammation and cell death, it is observed that cDPP3 is increased in the bloodstream, which could lead to degradation of cardiovascular peptides and hemodynamic instability. 

DPP4 is a serine protease localized in the membrane of cells while DPP3 is present in the cytosol. Like DPP3, it is a dipeptidyl aminopeptidase. Inhibition of DPP4 activity is of considerable interest for the therapy of type 2 diabetic patients. For now, there are no studies showing any relationship between DPP3 and DPP4. The number of DPP was conventionally added according to the time of its discovery (Prajapati et al. FEBS 2011). 

The manuscript was modified according to this comment

Introduction section, page 1, line 24: “Exact roles of DPP3 in the heart and especially in cardiac cells are not completely understood.”

2) Why the antibody inhibits DPP3 activity? Is the epitope close to the active site of the enzyme?

Response: The humanized DPP3-inhibiting antibody Procizumab binds to a conserved, surface exposed loop in proximity to the active site. Crystal structures of human DPP3 in apo and substrate-bound forms show that the enzyme undergoes a large conformational change between the open (apo) state and the closed (substrate-bound state). Since the epitope targeted by Procizumab remains exposed in the surface in the open and closed states, it is expected that the antibody binds the enzyme regardless of its conformation and sterically blocks the transition between DPP3 catalytic states, therefore, inhibiting enzyme activity. 

The manuscript was modified according to this comment.

Page 4, line 81: “Inhibition of DPP3 activity by PCZ

Generation, screening and development of humanized anti-DPP3 antibody PCZ was previously described [7]. In addition, the humanized DPP3-inhibiting antibody Procizumab binds to a conserved, surface exposed loop in proximity to the active site. Crystal structures of human DPP3 in apo and substrate-bound forms show that the enzyme undergoes a large conformational change between the open (apo) state and the closed (substrate-bound state). Since the epitope targeted by Procizumab remains exposed in the surface in the open and closed states, it is expected that the antibody binds the enzyme regardless of its conformation and sterically blocks the transition between DPP3 catalytic states, therefore, inhibiting enzyme activity.”

3) Why DPP3 activity was measured only in the plasma but not in the heart of the animals? Is DPP3 cleaved by a sheddase? Is this sheddase upregulated in sepsis?

Response: We thank the reviewer for this intriguing comment. Indeed, DPP3 activity was only measured in plasma since the antibody can only bind and inhibit DPP3 activity in plasma and not inside cells, e.g. cardiac cells. Although membrane-associated forms of DPP3 have been reported, we have not been able to detect membrane-bound DPP3 in a variety of tissues, such as heart, liver, kidney and lungs. To our understanding and based on a large body of literature, DPP3 is mainly an intracellular enzyme and, therefore, not exposed to the activity of sheddases. Nevertheless, we do agree that further studies are needed to specifically measure DPP3 activity and concentration in the heart of healthy and diseased animals.

The manuscript was modified in the limitation section according to this comment

Page 11, line 220: “Fourth, DPP3 activity was only measured in plasma since the antibody can only bind and inhibit DPP3 activity in plasma and not inside cells, e.g. cardiac cells. 

4) Very little is shown and discussed regarding the potential mechanisms by which DPP3 inhibition improves cardiac function in the setting of sepsis.

Response: Indeed, the potential mechanism of action by which DPP3 inhibition improves cardiac function is currently unknown in sepsis. This is a proof-of-concept study showing a role of DPP3 in myocardial dysfunction during sepsis. Cardiac and vascular haemodynamic measurements and further biological analyses should be performed in future preclinical studies to better understand the mechanism(s) of action of PCZ in the sepsis.

The manuscript was modified in the limitation section according to this comment.

Page 10, line 217: “Third, the mechanism of action of DPP3 and its inhibition by PCZ in septic cardiomyopathy is currently poorly understood.”

5) Was the lack of variation in heart rate among the three groups of rats expected?

Response: We thank the reviewer for this valuable comment. Indeed, the heart rate did not exceedingly vary during the first 120 min after PBS or PCZ injection. A decreasing trend in the PCZ group was, however, observed. We did observe an increase in stroke volume and cardiac output in this PCZ-treated group as specified in the manuscript (Page 8, line 164 : In contrast, heart rate (HR) and MBP measured at 120 minutes were similar in all CLP groups (Fig 2F and G)). 

6) MINOR

Please, replace angiotensin 2 by angiotensin II or Ang II.

Response: We thank the reviewer for this comment and we modified all the manuscript accordingly.

---

## [Decision Letter · Decision Letter 1]

10 Aug 2020

Inhibition of circulating dipeptidyl-peptidase 3 restores cardiac function in a sepsis-induced model in rats: a proof of concept study

PONE-D-20-10991R1

Dear Dr. Deniau,

We’re pleased to inform you that your manuscript has been judged scientifically suitable for publication and will be formally accepted for publication once it meets all outstanding technical requirements.

Kind regards,

Vincenzo Lionetti, M.D., PhD

Academic Editor

PLOS ONE

Additional Editor Comments (optional):

Reviewers' comments:

Reviewer's Responses to Questions

**Comments to the Author**

1. If the authors have adequately addressed your comments raised in a previous round of review and you feel that this manuscript is now acceptable for publication, you may indicate that here to bypass the “Comments to the Author” section, enter your conflict of interest statement in the “Confidential to Editor” section, and submit your "Accept" recommendation.

Reviewer #1: All comments have been addressed

Reviewer #2: All comments have been addressed

2. Is the manuscript technically sound, and do the data support the conclusions?

Reviewer #1: Yes

Reviewer #2: (No Response)

3. Has the statistical analysis been performed appropriately and rigorously? 

Reviewer #1: (No Response)

Reviewer #2: (No Response)

4. Have the authors made all data underlying the findings in their manuscript fully available?

Reviewer #1: (No Response)

Reviewer #2: (No Response)

5. Is the manuscript presented in an intelligible fashion and written in standard English?

Reviewer #1: (No Response)

Reviewer #2: (No Response)

6. Review Comments to the Author

Reviewer #1: (No Response)

Reviewer #2: (No Response)

7. PLOS authors have the option to publish the peer review history of their article (what does this mean?). If published, this will include your full peer review and any attached files.

Reviewer #1: **Yes: **Jean-Louis Vincent

Reviewer #2: No

---

## [Editor Report · Acceptance letter]

18 Aug 2020

PONE-D-20-10991R1 

Inhibition of circulating dipeptidyl-peptidase 3 restores cardiac function in a sepsis-induced model in rats: a proof of concept study 

Dear Dr. Deniau:

I'm pleased to inform you that your manuscript has been deemed suitable for publication in PLOS ONE. Congratulations! Your manuscript is now with our production department. 

Kind regards, 

on behalf of

Prof. Vincenzo Lionetti 

Academic Editor

PLOS ONE